# A Tenon's capsule/bulbar conjunctiva interface biomimetic to model fibrosis and local drug delivery

Katarzyna Kozdon[1⊕], Bruna Caridi[1⊕], Iheukwumere Duru[1¤], Daniel G. Ezra[1,2], James B. Phillips[3], Maryse Bailly[1]*

**1** UCL Institute of Ophthalmology, UCL, London, United Kingdom, **2** Moorfields Eye Hospital, London, United Kingdom, **3** Department of Pharmacology, UCL School of Pharmacy, UCL, London, United Kingdom

⊕ These authors contributed equally to this work.
¤ Current address: Warrington and Halton Teaching Hospitals NHS Foundation Trust, Warrington, United Kingdom
* m.bailly@ucl.ac.uk

**Data Availability Statement:** All relevant data are within the paper and its Supporting Information files.

## Abstract

Glaucoma filtration surgery is one of the most effective methods for lowering intraocular pressure in glaucoma. The surgery efficiently reduces intra-ocular pressure but the most common cause of failure is scarring at the incision site. This occurs in the conjunctiva/Tenon's capsule layer overlying the scleral coat of the eye. Currently used antimetabolite treatments to prevent post-surgical scarring are non-selective and are associated with potentially blinding side effects. Developing new treatments to target scarring requires both a better understanding of wound healing and scarring in the conjunctiva, and new means of delivering anti-scarring drugs locally and sustainably. By combining plastic compression of collagen gels with a soft collagen-based layer, we have developed a physiologically relevant model of the sub-epithelial bulbar conjunctiva/Tenon's capsule interface, which allows a more holistic approach to the understanding of subconjunctival tissue behaviour and local drug delivery. The biomimetic tissue hosts both primary human conjunctival fibroblasts and an immune component in the form of macrophages, morphologically and structurally mimicking the mechanical proprieties and contraction kinetics of *ex vivo* porcine conjunctiva. We show that our model is suitable for the screening of drugs targeting scarring and/or inflammation, and amenable to the study of local drug delivery devices that can be inserted in between the two layers of the biomimetic. We propose that this multicellular-bilayer engineered tissue will be useful to study complex biological aspects of scarring and fibrosis, including the role of inflammation, with potentially significant implications for the management of scarring following glaucoma filtration surgery and other anterior ocular segment scarring conditions. Crucially, it uniquely allows the evaluation of new means of local drug delivery within a physiologically relevant tissue mimetic, mimicking intraoperative drug delivery *in vivo*.

**Funding:** This work was supported by the Special Trustees of Moorfields Eye Hospital (now Moorfields Eye Charity; https://moorfieldseyecharity.org.uk/; grant ST 12 11 B to MB and DGE). DGE acknowledges financial support from the Department of Health through the award made by the National Institute for Health Research to Moorfields Eye Hospital NHS Foundation Trust and UCL Institute of Ophthalmology for a Biomedical Research Centre for Ophthalmology (https://www.moorfieldsbrc.nihr.ac.uk/). The views expressed in this publication are those of the authors and not necessarily those of the Department of Health. The funders had no role in study design, data collection and analysis, decision to publish, or preparation of the manuscript.

**Competing interests:** The authors have declared that no competing interests exist.

## Introduction

Ocular fibrosis is a pathological feature of a number of sight-threatening diseases, such as trachoma, keratoconjunctivitis, and glaucoma, and a major cause of post-surgical complications and treatment failure, leading to blindness in millions of people. Despite recent improvements, the management of complications due to tissue contraction and scarring is still not satisfactory. With only a few anti-fibrotic treatments currently licensed for specific diseases, this is one of the largest unmet medical needs in ophthalmology and medicine [1]. For glaucoma alone, over 70 million people worldwide are affected, including 500,000 in the UK [2]. Glaucoma is most commonly treated with eye drops or laser aimed at reducing IOP. However, when adequate pressure control can no longer be controlled using eye drops, surgery is required, most often trabeculectomy, on a worldwide scale.

Trabeculectomy involves making an incision into the sclera and creating a canal to allow aqueous humour to drain into the sub-Tenon's/conjunctival space, forming a "bleb", and lowering pressure. Successful surgery yields a permanent draining bleb and intraocular pressure (IOP) control [3]. However, scarring often develops at the draining site, eventually leading to bleb failure. Up to 40% of patients may need adjunct topical eye drop treatment or additional surgery to suitably control IOP [1, 3–5]. In lower income countries, surgery is often the only practical therapy for glaucoma. Furthermore, scarring results in suboptimal lowering of the intraocular pressure, which is associated with a higher final pressure and poorer long term outcomes of surgery. Consequently, improving surgical success is critical in such settings to improve outcomes, and will be even more so in the coming decades, as over 110 M people are predicted to have glaucoma by 2040, most significantly in Africa and Asia [2].

Anti-metabolite agents such as 5 fluorouracil or mitomycin C have been used successfully to reduce post-operative scarring in glaucoma, but they have side effects, including tissue damage and infection, which can lead to blindness [5–7]. In addition, inflammation is a major driver of postoperative fibrosis, and a number of adjunctive treatments can be used to modulate the inflammatory response after surgery including steroids and non steroidal anti-inflammatory drugs (NSAIDs), reducing the risk of scarring [7]. However, evaluations of such adjunctive treatments have yielded variable results, and often they are not suitable for use in low-income countries due to long postoperative follow up [8]. A better understanding of the fibrotic response in the conjunctiva, and the response to therapy, is thus critical to develop safer and more efficient treatments to prevent postoperative scarring.

A major reason for the unmet demand for clinical treatment for fibrosis is the lack of a reliable *in vitro* model of conjunctival fibrosis. Although some recently developed models have included animal tissue, and/or some form of engineered 3D tissue [9–11], the most commonly used *in vitro* model to study contraction in the conjunctiva and screen potential anti-scarring treatment is the fibroblast-populated collagen lattice [6, 12]. While crucial in helping the development of the current anti-scarring regimen used in glaucoma filtration surgery [6], it has since been of limited use to the development of less toxic anti-scarring drugs and/or new delivery systems, as it uses very soft, loose gels that do not suitably represent human tissue composition and stiffness [9, 13]. In addition, most *in vitro* models of conjunctival fibrosis use simple settings of fibroblasts embedded in matrix, without the additional inflammatory component often found at the surgery site. As a result, most drug development studies are undertaken in animal models—rabbit and more recently mouse for glaucoma [14–17]—with significant ethical and economic constraints and often little information on the complex biological processes involved. Similarly, while models of drug diffusion and pharmacokinetics in the eye have been described, they often involve placing the drug-containing device in liquid medium to measure release, which does not suitably mimic drug diffusion in physiological tissues [18]. Indeed, the

tissue itself may have a significant impact on flow, particularly when microimplants are used in different positions [19].

In this study, we have engineered an *in vitro* model of conjunctiva, which combines many desirable features of the current models: it allows for control and molecular manipulation of both cells and extracellular environment, and can host patient-derived cells such as fibroblasts and macrophages to model normal physiology and inflammatory conditions. The engineered construct contraction profile mimics *ex vivo* tissue contraction, making it suitable for examining such aspects of scarring and fibrosis as cell motility, matrix remodelling and degradation—effectively acting as a conjunctiva biomimetic. Importantly, our tissue constructs, by reproducing the clinically relevant features of the subconjunctival tissue, i.e. the layered structure with differences in tissue stiffness and the presence of an immune component, allow for the assessment of local drug delivery to the Tenon's capsule-bulbar conjunctiva interface, effectively mimicking intraoperative drug delivery.

## Methods

### Cells and tissues

Primary human Tenon's capsule fibroblasts (HTF0748-2; HTF1785R; HTF9154) were isolated from donor tissue with written informed consent for research use in accordance with the tenets of the Declaration of Helsinki and local ethics approval as previously described [9] (ETR reference 10/H0106/57-2011ETR18, approved 18/6/2012 by the Eye Tissue Repository Internal Ethics Committee of the Moorfields Eye Hospital Eye Tissue Repository). The cells were cultured in DMEM supplemented with 10% foetal bovine serum (FBS, Sigma-Aldrich, Gillingham, UK), 100 IU/ml penicillin, 100μg/ml streptomycin (Life Technologies, Thermo Fischer Scientific, Paisley, UK).

Human monocyte cell line U937 was cultured in RPMI-1640 medium (Sigma-Aldrich, Gillingham, UK) supplemented with 10% FBS, 100 IU/ml penicillin and 100μg/ml streptomycin [20]. The monocytes were differentiated into macrophages by adding 1 μg/ml of phorbol myristate acetate (PMA, Promega Corporation, WI, USA) in RPMI-1640 for 36 hours, then left to recover in RPMI-1640 for 36 hours. For ex-vivo conjunctiva tissue measurements, freshly obtained porcine eyes were processed, and tissue was cultured as described previously [9].

### Compressed collagen gels

Compressed gels were made by mixing 1900 μl of rat tail collagen type I in acetic acid (2.1 mg/ml, First Link UK, Wolverhampton) with 330 μl of concentrated medium (700 μl of 10x DMEM [Sigma-Aldrich, Gillingham, UK], 70 μl of L-glutamine [Gibco, Life Technologies, Paisley, UK], and 180 μl of 7.5% sodium bicarbonate [Sigma-Aldrich, Gillingham, UK]). The pH was adjusted to 7–7.5 with NaOH (100 μl of a 2M solution). Cells were resuspended in FBS and added to the mix. 300 μl of the gel was dispensed per one custom-made 10 mm diameter steel ring. Gels were incubated for 40 minutes at 37˚C, and then sandwiched between two pieces of nylon mesh placed on Whatman blotting paper (GE Healthcare UK Limited, Little Chalfont), then covered with a glass slide and compressed with 51 g weights (Thermo Fischer Scientific, Paisley, UK) for 5 minutes. Following compression, the nylon mesh was removed, and the resulting compressed gels were transferred to 12 well plates, where 2 ml of medium was added to each well.

### Standard collagen gels

Collagen matrices were prepared according to a modified version of a previously published protocol [21]. Briefly, cells were trypsinised, counted and resuspended in FBS, and were seeded at a density of 74 cells/μl in a 1.5 mg/ml collagen type I gel (First Link UK, Wolverhampton) supplemented as described by Dahlmann-Noor et al. [13].

## Bilayer gels

Compressed gels were prepared as described above, transferred to MatTek dishes (MatTek Corporation, MA, USA), and standard collagen gels were cast on the top of the compressed gels.

## Mechanical properties measurements

Elastic modulus was measured using a dynamic biomechanical analyser (Bose Elctroforce 3200, TA Instruments, using WinTest 7 software) at room temperature in uniaxial tension mode. For the test, larger biomimetic tissues were prepared, using 12-well plate and upscaled. The porcine tissues and biomimetic samples were cut using a circular cutter with the help of a graph rectangular base into small mirrored hourglass-shaped pieces, with enlarged tips for gripping (Fig 8A). The reduction in the cross-sectional area of central section (neck) relative to that of the remainder of the specimen allowed for localized deformation and failure of the specimen within the gauge section.

The specimens' parameters (width and thickness) were first evaluated in ImageJ (http://imagej.nih.gov/ij/) from photos taken with KSV's CAM 200 (CCD firewire camera (512x480) with telecentric zoom optics) for the thickness, and Huawei Ascend P7 Camera (13megapixel, FullHD) for the width, using the average of 3 measurements for each.

The samples were then clamped with the help of an appropriate piece of paper and a tensile force was applied at an extension rate of 0.5 mm/s.

The stress-strain curve was plotted using the formula

$$E = \left(\frac{F}{A}\right)\left(\frac{L}{\Delta L}\right)$$

where $A$ is the unstressed cross-sectional area calculated by $A = thickness^* width$, F is the force, L is the unstressed length, and $\Delta L$ is the change in length.

The slope of the stress–strain curve in the linear deformation region was calculated to obtain the elastic modulus of the samples.

## Macroscopic contraction and pharmacological treatments

Compressed gels and porcine conjunctiva were incubated in 12 well plates in DMEM with 10% FBS, supplemented with drugs (test) or drug vehicles (control), and contraction was monitored for 21 days [9]. Doxycycline hyclate (Sigma-Aldrich, Gillingham, UK) was dissolved in water and used at the final concentration of 416 μM (as described in Li et al. [19]), with the medium replaced with a fresh solution once a week. Ehop-016 and NSC23766 (both from Tocris Biosciences, Abingdon, UK) were resuspended in dimethyl sulfoxide, and added for 24 hours every seven days at final concentrations of 10 μM and 50 μM respectively.

## Confocal imaging

Samples were fixed in 3.7% paraformaldehyde in phosphate-buffered saline (PBS), permeabilised with 0.5% Triton X-100, washed with 0.1M glycine and incubated with rhodamine-phalloidin (1:20, Molecular Probes, Thermo Fischer Scientific, Paisley, UK) in Tris-buffered saline (TBS) with addition of 2% FBS, followed by washing and mounting with n-propyl gallate (6g/l, Sigma-Aldrich, Gillingham, UK) in 50% glycerol in TBS. Images were taken using Zeiss Axiovert S100/Biorad Radiance 2000 confocal microscope (Bio-Rad Laboratories Ltd., Hemel Hampstead, UK), objectives 20x, method Red/CRM/Trans, and reconstructed using Volocity 6.1.1. Software (PerkinElmer, MA, USA).

### Second harmonic generation

Collagen fibres in standard and compressed gels were imaged live and after 10 minute fixation in 3.7% paraformaldehyde. A Leica SP8 with Chameleon System, Vision II, Integrated Pre-compensation, 80 Mhz Ti-Saphire laser (Coherent UK Ltd) was used at 880nm pump wavelength to take second harmonic generation images as described previously [22]. Samples were imaged with 25x 0.95NA water dipping objective at 1024 x 1024 pixels resolution at 430–450 nm bandwidth with the external, non-descanned, Hybrid detectors.

### Cell viability assays

The Live/Dead cell viability assay (Thermo Fischer Scientific, Paisley, UK) was used according to the manufacturer's instructions. Live (green) and dead (red) cells were visualised using Zeiss Axiovert S100/Biorad Radiance 2000 confocal microscope as described above.

Cell toxicity was assessed in the medium on a weekly basis using Pierce LDH Cytotoxicity Assay kit (Thermo Fischer Scientific, Paisley, UK) according to the manufacturer's instructions. Values were normalised to readouts obtained by measuring LDH in medium alone (0% dead cells) and samples where cells were lysed with RIPA buffer (150 mM sodium chloride, 1% Trition X-100, 0.5% sodium deoxycholate, 0.1% sodium dodecyl sulphate, 50 mM Tris) for 24 hours (100% dead cells, confirmed with Live/Dead assay). For experiments involving a drug treatment, background signal (cell medium with a drug) was measured separately for each drug, and subtracted from cell culture measurements.

### Metabolic activity assay

Samples were incubated at 37˚C in the presence of 10% Alamar Blue stain (Thermo Fischer Scientific, Paisley, UK) in standard medium for 1 hour. 100 μl of solution was transferred to a well in a 96-well plate (Thermo Fischer Scientific, Paisley, UK), and the fluorescence was measured using excitation and emission wavelengths of 544 nm and 590 nm respectively (Fluostar Optima, BMG Labtech, Germany).

### Protein content measurement in gels

Total gel protein content was measured as a readout for collagen degradation [12]. Gels were fixed in 4% formaldehyde for 30 minutes then stained in Coomassie Blue for 30 minutes. The dye was extracted from the gels with 70% ethanol for 1 hour, and samples were measured for their absorbance at 550 nm (Fluostar Optima; BMG Biotech, Cary, NC).

### Statistical analysis

All experiments were performed in triplicate, and with at least three independent repeats. Figures show mean and SEM of the three repeats. Statistical analysis was performed using ANOVA and one-tailed student's t-test where appropriate.

## Results

### Fibroblast-embedded plastic compressed collagen gels mimic conjunctival stroma morphological features and contractile properties

Plastic compression of collagen hydrogels has been used previously to generate tissue-like structures [23, 24] including a cell-free basal support for conjunctival epithelium [11]. Using similar techniques, we generated tissue-like structures by embedding primary human conjunctival fibroblasts within the compressed hydrogels. We found that a cell density of 200 cells/μl

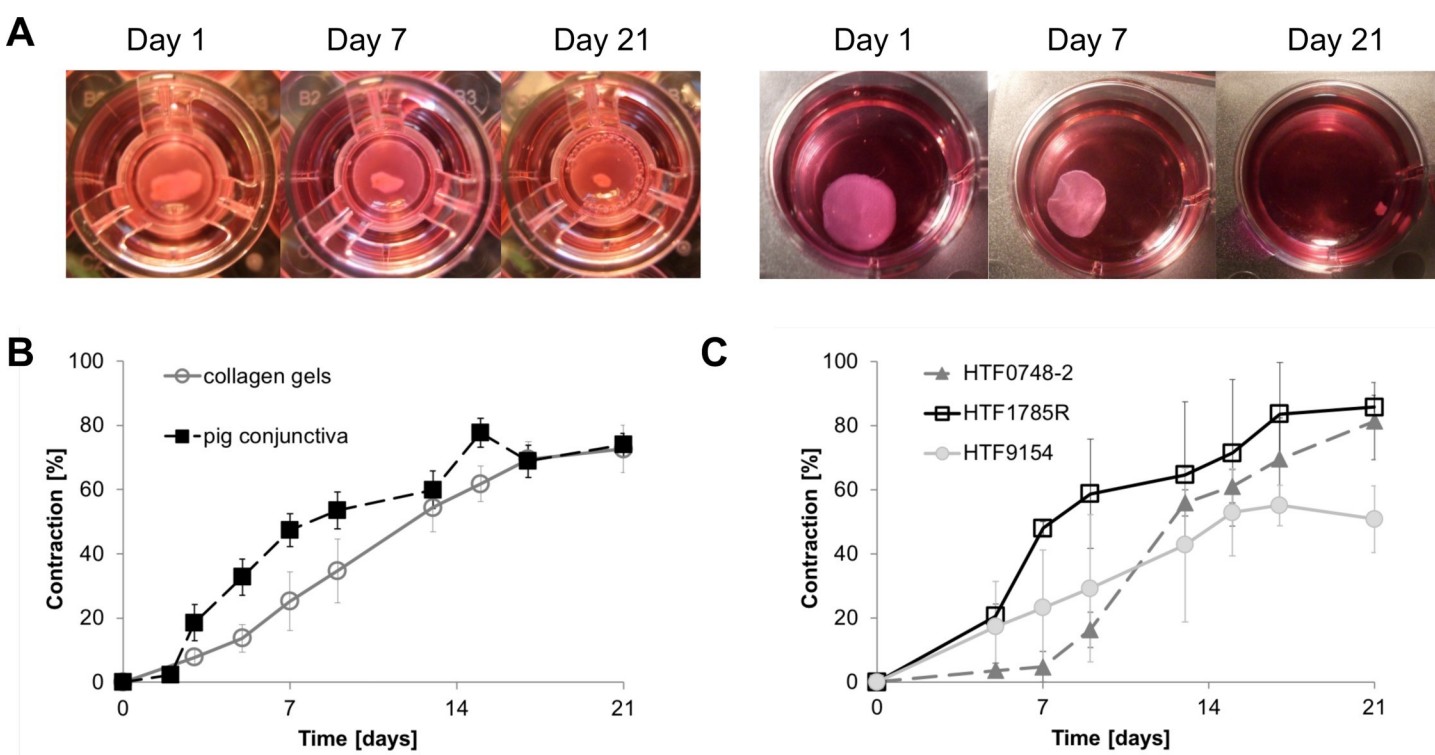

**Fig 1. Cell-seeded compressed hydrogels contract with physiological kinetics.** Porcine conjunctival fragments and plastic-compressed gels containing primary human conjunctival fibroblasts (HTF0748-2; HTF1785R; HTF9154) were placed in 12 well-plates in medium with 10% serum, and contraction was monitored at least twice a week for 3 weeks. A) Porcine conjunctiva (left) and compressed collagen gels (right) on days 1, 7 and 21 in culture. B) Contraction kinetics of porcine conjunctiva (n = 6, 8–12 pieces per time point), and collagen gels seeded with human conjunctival fibroblasts (n = 6, pooled data for all 3 cell lines). C) Individual contraction kinetics for compressed hydrogels made with each primary fibroblast line (n = 2 per cell line, 3 gels per repeat). Graphs show mean ± SEM.

collagen solution (before compression, S1 Fig) yielded tissues with similar cell densities as porcine [9, 25] and human [13, 26, 27] conjunctiva. These engineered tissues contracted in the presence of 10% serum, with overall kinetics similar to those measured for intact conjunctival fragments in our *ex vivo* model of conjunctival scarring using porcine tissue [9] (Fig 1A and 1B), and with reproducibly comparable kinetics for the 3 different primary cell cultures used (Fig 1C). Microscopic analysis of the tissues revealed similar features for the pig conjunctiva and the cell-seeded compressed gels, with comparable cell densities and a compact matrix structure [9, 13, 25] (Fig 2 and S2 Fig). Cell viability and metabolic activity in gels were analysed using LDH and Live/Dead assays, and Alamar Blue assay respectively, with measurements taken on days 7, 14 and 21 in culture. Compressed collagen gels maintained a high cellularity throughout the 3 weeks of contraction, with low levels of cell death as measured by LDH (maximum 25%; Fig 3A), consistent with that measured in the pig conjunctiva under similar conditions in our *ex vivo* assay [9]. Similarly, the Live/Dead staining performed on day 21 confirmed a high cellularity with a minimal number of dead cells (Fig 3B), as seen in intact conjunctiva fragments after 28 days in culture. Cells maintained metabolic activity throughout the three-week period, again, suggesting that compressed gels are a viable tissue mimic (Fig 3C).

### Fibroblast-embedded plastic compressed collagen gels as a model to screen anti-scarring drugs

To determine whether our engineered conjunctiva tissues behaved as intact conjunctiva with respect to treatment with potential anti-scarring drugs, we evaluated the effect of small GTPase

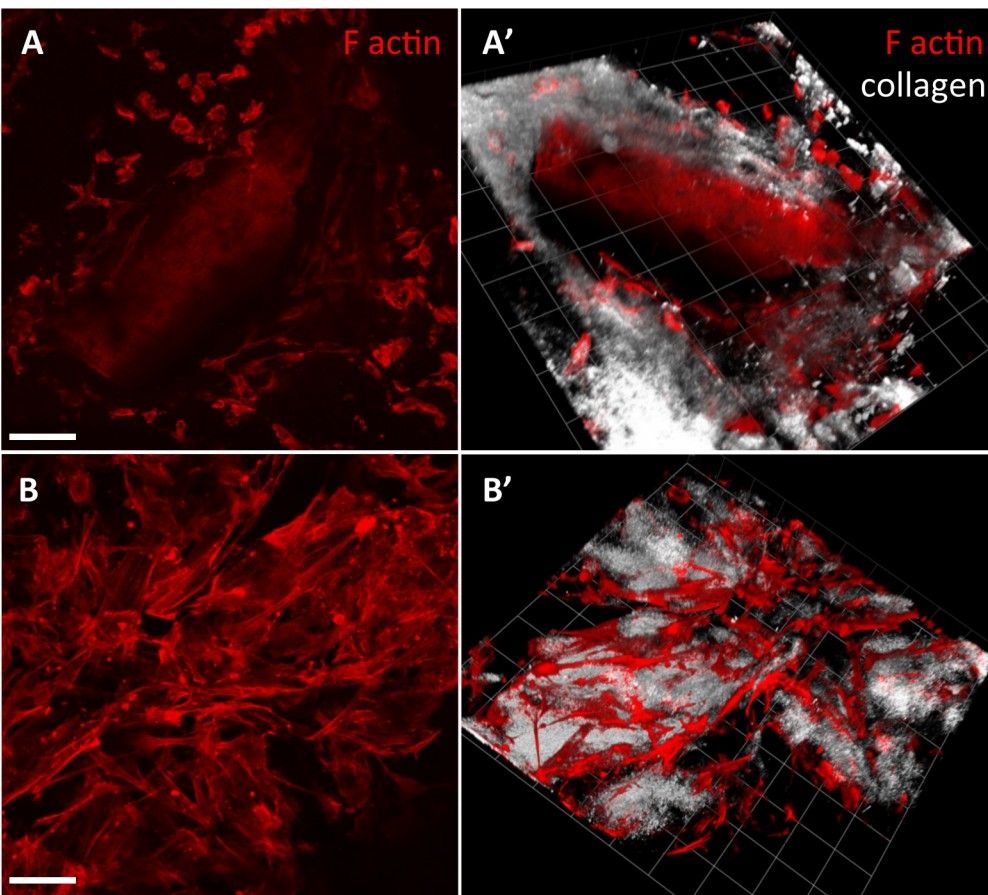

**Fig 2. Cell-seeded compressed hydrogels and porcine conjunctiva display a similar architecture.** Porcine conjunctival fragments and plastic compressed gels containing primary human conjunctival fibroblasts (HTF9154) were placed in medium with 10% serum and left to contract for 10 days. The tissues were then fixed, stained for F-actin using Rhodamine-labelled phalloidin, and imaged using confocal microscopy using fluorescence (F-actin, red) and reflection microscopy (collagen matrix, white). Shown are representative extended views (maximum intensity projection; A and B) and 3D reconstruction (A' and B') of porcine conjunctiva (A and A') and compressed collagen tissues (B, B'). Scale bar, 90 μm. 3D view grid: one unit, 62.33 μm.

inhibitors Ehop-016 and NSC23766, as well as doxycycline, all of which significantly inhibited contraction in fibroblast-populated lattices and *ex-vivo* conjunctiva contraction ([9, 12, 28]; S3 Fig). Doxycycline [416 μM] was maintained throughout the three weeks period (refreshed weekly), whereas Ehop-016 [10 μM] and NSC23766 [50 μM] were added for 24 hours only once a week, and the medium was then replaced with fresh, drug-free, medium as per previous work [12]. Despite previously reducing the contraction of *ex-vivo* conjunctival fragments [12], NSC23766 only weakly inhibited engineered tissue contraction over 21 days (Fig 4A); examination of NSC23766 in non-compressed collagen gels confirmed its transient effect on contraction (S4 Fig, as shown previously [12]). On the other hand, both Ehop-016 (Fig 4B) and doxycycline (Fig 4C) dramatically inhibited contraction up to day 21. None of the drugs showed signs of significant toxicity in the LDH assay (Fig 4D), with LDH levels being comparable to or lower than in the control. The Alamar Blue assay (Fig 4E) showed that Ehop-016 significantly reduced metabolic activity; NSC23766 and doxycycline treatments only mildly affected the metabolic activity. Consistent with our previous work [12], the effect (or lack of thereof) of the drugs on contraction was matched by their effect on matrix degradation as

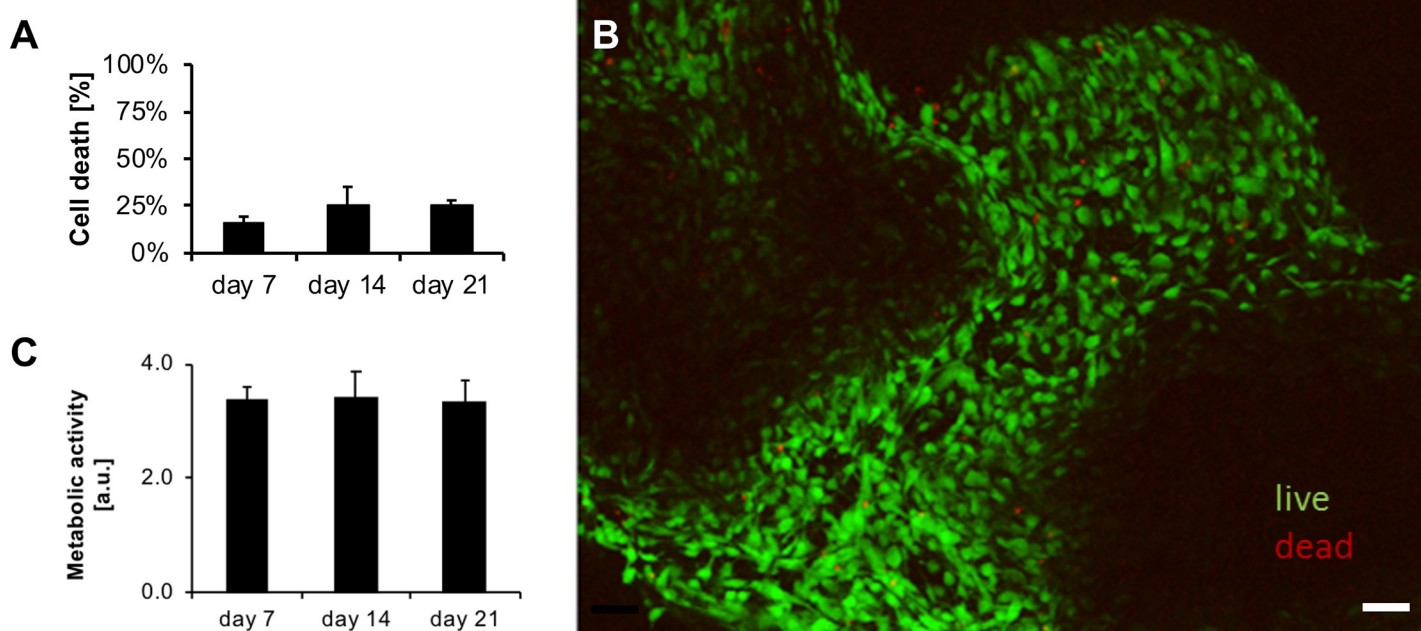

**Fig 3. Compressed collagen gels maintain a high cellularity throughout contraction.** Plastic compressed gels containing primary human conjunctival fibroblasts (HTF0748-2) were placed in medium with 10% serum and the gels were left to contract for 3 weeks. Cell viability and proliferation were assessed on day 7, 14 and 21, using LDH and Alamar Blue assays respectively, and confirmed visually using a Live/Dead assay at day 21. A) LDH assay: cell viability is expressed with reference to the 100% death control (cells lysed with RIPA buffer). Cell death was minimal throughout the 3 weeks of contraction. B) Live/Dead assay: 3-week-old tissues were well populated, with almost 100% viable cells (green) and only a few dead (red) cells. Scale bar, 200 um. C) Alamar Blue assay: there was no significant change in metabolic activity during contraction. Graphs show mean ± SEM for 3 independent experiments, with 2 replicates each.

measured by a global protein measurement using Coomassie Blue, with both doxycycline- and Ehop-016-treated samples displaying protein levels significantly higher than the control, suggesting that matrix degradation was successfully prevented by the treatments (Fig 4F).

## Macrophages can be incorporated in compressed collagen tissues with maintenance of the contraction potential and structure

Immune cells, and particularly macrophages [29], are present in significant amounts in normal conjunctiva (S2 Fig and [25]) and their numbers can increase dramatically during inflammation and fibrosis/scarring [26, 30, 31]. In order to mimic the normal inflammatory component in our conjunctiva model, we incorporated U937-derived human macrophages at a 2:1 fibroblast to macrophage ratio, which is consistent with the estimated ratio reported in human conjunctiva [26, 27]. We have shown previously that U937-derived macrophages are fully differentiated, and functionally interact with both normal and fibrotic fibroblasts in our standard 3D contraction model [20]. The total cell density was adjusted to 240 cells/µl to maintain a physiologically relevant tissue structure. The resulting bi-cellular compressed tissues displayed contraction profiles (Fig 5A) and tissue architecture (Fig 5B) similar to those of porcine tissue (compare to Figs 1B and 2A).

## Reconstructing the bulbar conjunctiva/Tenon's capsule interface

Intraoperative drug delivery is one of the key ways of preventing and treating ocular fibrosis, yet *in vitro* models for local drug delivery <u>within</u> tissue-like structures are lacking. Fibroblast-populated collagen lattices have long been used as a surrogate model for tissue contraction to screen potential anti-scarring treatments [32], especially in the context of preventing post-

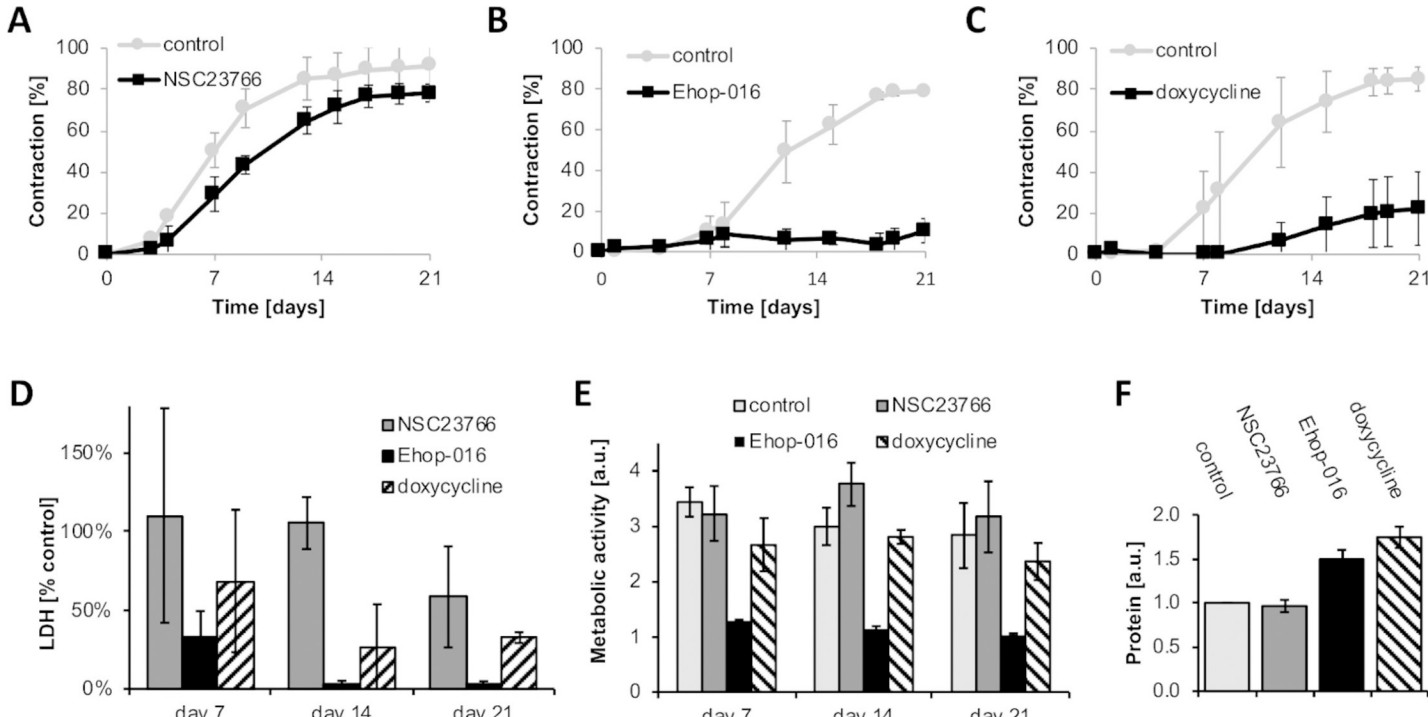

**Fig 4. Using engineered conjunctiva stroma to screen potential anti-scarring drugs.** Plastic compressed gels containing primary human conjunctival fibroblasts (HTF0748-2) were placed in medium with 10% serum with/without anti-scarring drugs, and monitored over the period of 21 days. A, B, C) Contraction kinetics of gels exposed to NSC23766 [50 µM], Ehop-016 [10 µM] and doxycycline [416 µM] respectively for 24hrs, and their corresponding controls without drug. D) Toxicity assay: on day 7, 14, and 21, cell death was evaluated using the LDH assay. Graph shows LDH levels normalised to untreated control. E) Alamar blue assay: on day 7, 14, and 21 the metabolic activity in the gels was examined using the Alamar Blue assay. F) Coomassie Blue assay: on day 21, the amount of protein in the gels was examined using Coomassie Blue assay. Matrix degradation was reduced in gels exposed to Ehop-016 and doxycycline hyclate. Graphs show mean ± SEM for 3 independent experiments, each in triplicate.

surgical scarring after glaucoma filtration surgery [12, 33, 34], using bulbar conjunctiva and/or Tenon's capsule fibroblasts [1, 35, 36]. However, these have been of limited use to the development of less toxic anti-scarring drugs and/or new delivery systems, as the soft, loose gels do not suitably mimic human tissue composition and stiffness, and are not readily amenable to

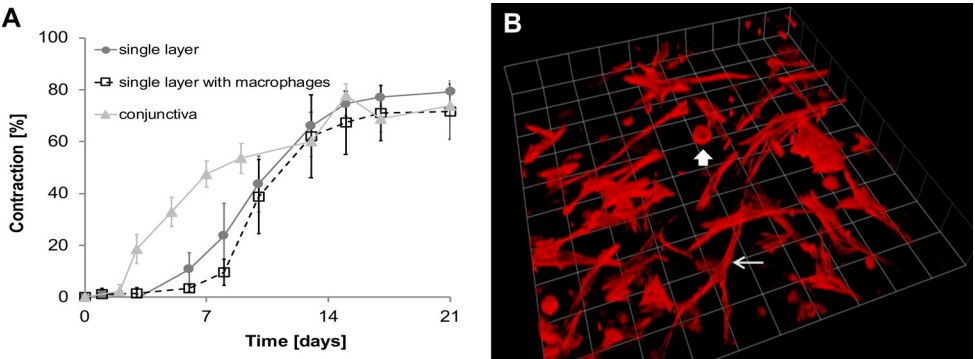

**Fig 5. Macrophages can be successfully incorporated in compressed collagen gels with maintenance of the contraction potential and structure.** A) Contraction kinetics of compressed gels with/without macrophages. Compressed gels with the addition of macrophages maintain a tissue-like contraction pattern. Porcine conjunctiva contraction from Fig 1B was replotted for reference. B) 3D reconstruction of compressed gel with fibroblasts (thin arrow) and macrophages (thick arrow) on day 21. 3D grid: 1 unit = 62.45 µm.

incorporation of slow release drug delivery devices. Having developed a suitable sub-epithelial conjunctiva stroma mimic, we attempted to reconstruct the bulbar conjunctiva/Tenon's capsule interface by combining compressed tissues as the subconjunctiva layer and soft collagen gels as the Tenon's capsule layer, as clinically, during surgery, the Tenon's capsule appears as a loose, soft tissue, while the overlaying subconjunctival and conjunctival tissues feel like a more discrete and stiffer layer (Prof. Sir Peng T. Khaw, personal communication). The stable bilayer tissue model displays a clear dual structure (Fig 6A and 6B). Using standard confocal reflection microscopy, the compressed layer is visualised as a dense compact tissue, while the top layer is much looser and not resolved at low magnification (Fig 6A, matrix coloured white). This pattern was confirmed by Second Harmonic Generation (SHG) microscopy, revealing the dense collagen layer at the bottom, and cells seemingly floating above (Fig 6B). Contrast enhancement reveals the cells are indeed embedded in a light hazy structure (Fig 6C). Higher magnification of a "floating" cell within the haze confirmed the collagen nature of the top layer, revealing a dense matrix of thin collagen fibres around the cells visible by confocal refection and SHG (Fig 6D and 6D', respectively). A full biomimetic can be achieved upon addition of macrophages to both layers (Fig 6E), while retaining physiological contraction kinetics (Fig 6F).

We used a dynamic biomechanical analyser to analyse the mechanical properties of the bilayer tissue model (Fig 7). The elastic modulus was measured at room temperature in uniaxial tension mode with a stretching force at an extension rate of 0.5 mm/s until sample rupture (Fig 7A and 7B). The stress-strain curve was plotted and the slope of the curve in the linear deformation region was calculated to obtain the elastic modulus of the samples. Stress-strain curves were obtained for porcine tissue (bulbar conjunctiva/Tenon's capsule fresh fragments, Fig 7C) and engineered tissue model (Fig 7C'). The porcine fragments displayed a low elastic modulus of 0.58±0.15 kPa (Fig 7D), consistent with the loose organisation of the tissue [25]. The conjunctiva biomimetic showed a higher elastic modulus (7.19±1.15 kPa), consistent with a more structured morphology [11], yet within the range of elastic modulus values for connective tissues [37].

Finally, we examined the suitability of our model for testing local drug delivery, in particular intraoperative drug delivery at the Tenon's capsule/bulbar conjunctiva interface. We used acellular compressed collagen gels soaked in PBS (control) and doxycycline as sample drug delivery vehicles, and inserted them between the engineered tissue layers. For illustrative purposes, an engineered tissue bilayer with sub-layers of different sizes and with a haematoxylin-coloured drug delivery insert is shown in Fig 8A. Control inserts did not affect the contraction profile of the model, while doxycycline-soaked inserts reduced contraction (Fig 8B). These results show that a drug-delivery vehicle can be inserted at the interface of the two modelled tissue types and that it remains inside the bilayer over the period of the experiment without mechanically affecting the contraction.

## Discussion

Despite recent improvements, the management of complications due to tissue contraction and scarring is still not satisfactory and, with only a few anti-fibrotic treatments currently licensed for specific diseases, this is one of the largest unmet medical needs in ophthalmology and medicine.

Postoperative scarring is the main cause of surgery failure in glaucoma and trachoma, two of the most common blinding diseases in the world. Here we describe an engineered 3D tissue model specifically designed to monitor mechanical and biological aspects of scarring and fibrosis in the conjunctiva, and evaluate potential new local treatments in a physiologically relevant context. Our engineered tissue emulates the human bulbar conjunctiva/Tenon's capsule

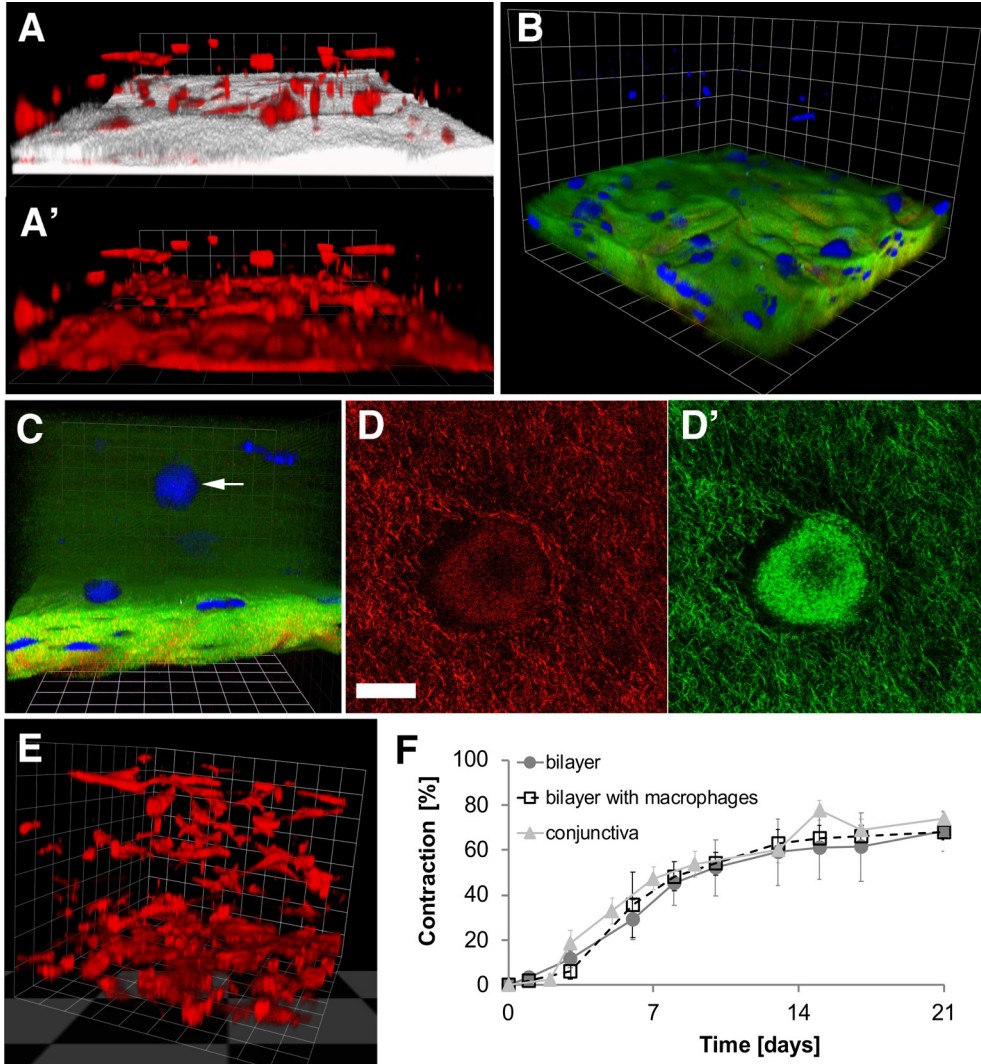

**Fig 6. Heterogenous bilayers mimic bulbar conjunctiva/Tenon capsule interface.** A) 3D confocal microscopy image of the interface of the two layers on day 21, stained for F-actin using Rhodamine-labelled phalloidin, and imaged using confocal microscopy using fluorescence (F-actin, red) and reflection microscopy (collagen matrix, white). Dense collagen layer can be seen in compressed gel (bottom, white), whereas the more hydrated uncompressed layer of collagen is invisible in the image (top, only "floating" cells are visible). 1 grid unit = 62.45 μm. A') Compressed gels retain a visibly higher cell density. B) 3D images of a bilayer gel with HTF and macrophages taken at day 25 using second harmonic generation microscopy. Green: collagen signal, blue: autofluorescent cells. The image shows low-density cell layout/loose collagen matrix in standard gel layer (top layer) and high-density layout/dense collagen matrix in compressed gel (bottom layer). 1 grid unit = 59.39 μm. C) 3D Images of a bilayer gel taken at day 1 using second harmonic generation microscopy. Green: collagen signal; blue: autofluorescent cells (HTF0748-2, selected cell indicated with an arrow); 1 grid unit = 33.58 μm. D) Higher magnification of the same cell imaged using confocal reflection and D') SHG microscopy, respectively, reveals the collagen network around the cell Scale bar = 33 μm. E) 3D confocal microscopy image of a bilayer gel with fibroblasts and macrophages at day 25; 1 grid unit = 62.45 μm. F) Contraction pattern of bilayer gels with human conjunctival fibroblasts and with/without macrophages (n = 3, mean± SEM). Porcine conjunctiva contraction from Fig 1B was replotted for reference.

interface, including biomechanical and biochemical features, as well as cell-cell and cell-matrix interactions and the potential role of inflammation.

Although a number of in vitro 3D/engineered models of glaucoma pathophysiology have emerged in recent years, most address the biology of the trabecular meshwork and regulation

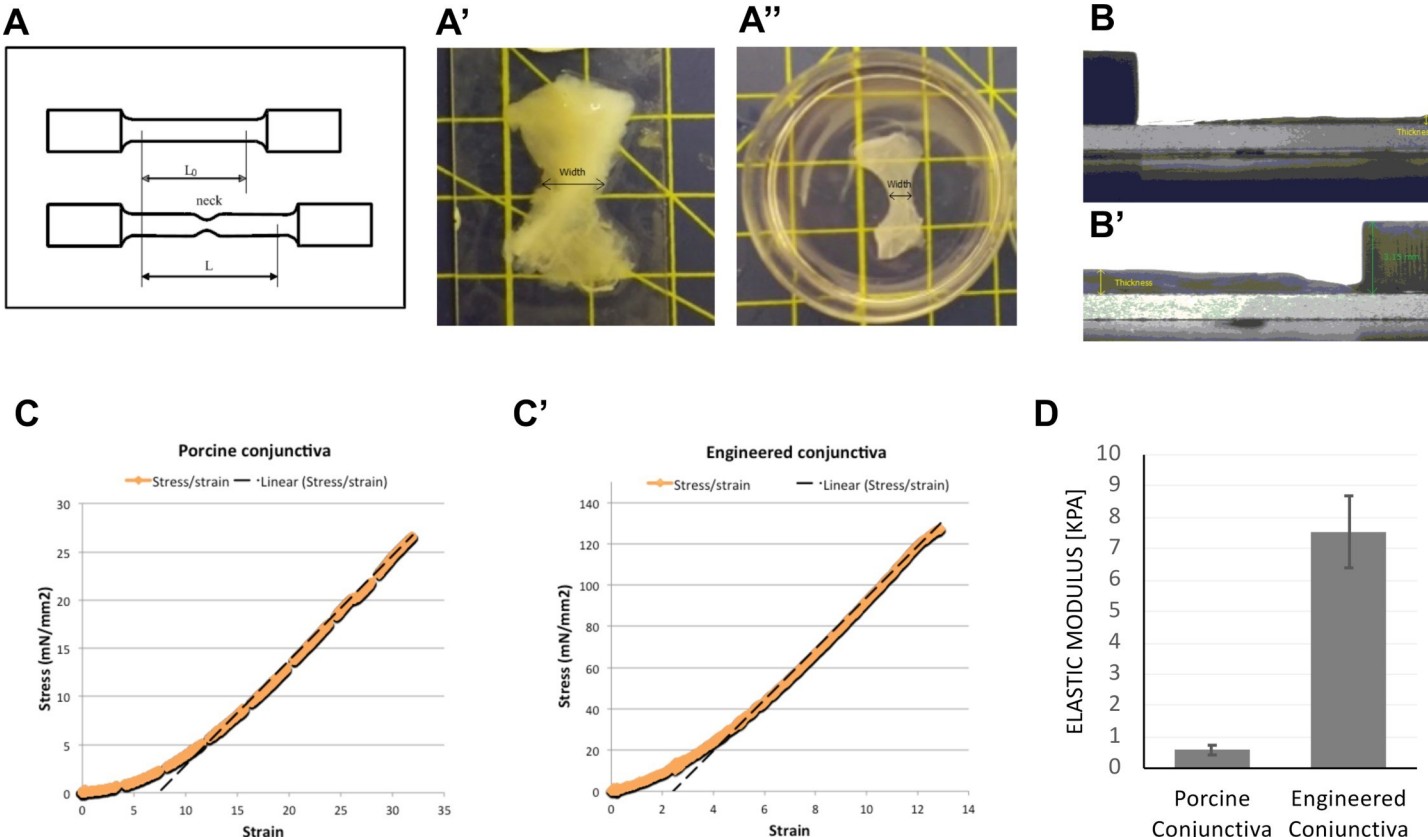

**Fig 7. Mechanical properties of engineered biomimetic and porcine conjunctiva.** A) Blueprint of a standard hourglass-shaped specimen for tensile test. A') Porcine conjunctiva was dissected to approximate the blueprint and its mechanical properties were measured within 24 hours of animal sacrifice. A") Engineered conjunctiva was created as previously, then cut to match the blueprint and its mechanical properties were measured within 24 hours. B, B') Porcine and engineered conjunctiva seen sidewise, with a coin as a reference for the thickness measurements. C, C') The typical stress-strain curve of the porcine and engineered conjunctiva obtained using a dynamic biomechanical analyser at room temperature in uniaxial tension mode with a stretching force at an extension rate of 0.5 mm/s until sample rupture. The slope of the curve in the linear deformation region was calculated to obtain the elastic modulus of the samples. D) Elastic modulus [kPa] as measured by dynamic biomechanical analysis for porcine conjunctiva and engineered conjunctiva. Graph shows mean and SEM for 8 independent experiments.

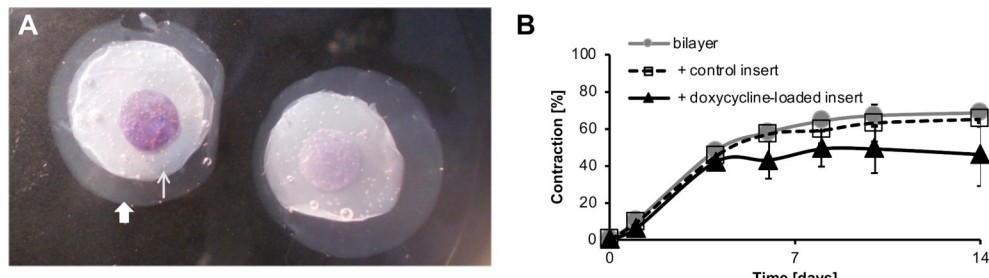

**Fig 8. Bilayer gels can be used to test local drug delivery.** A) Representative image of cell-seeded bilayer gel with a hydrated (thin arrow) and compressed (thick arrow) layer, and an acellular insert in between the two layers. For illustrative purpose, the two cell-seeded layers were made to differ in size and the insert was coloured with haematoxylin. B) Contraction kinetics of bilayer gels without an insert, with a control insert and with an insert soaked in a10 mM doxycycline solution. Doxycycline-soaked insert inhibited contraction, whereas control insert has no effect on the contraction pattern of the bilayer. N = 3, mean and SEM.

of intra-ocular pressure, or how this affects retinal ganglion cells [38–40]. Nevertheless, such models have demonstrated that 3D cultures are better at reproducing the pathophysiology of the disease, and better mimic *in vivo* behaviour, for example in terms of the cell responses to stress [40]. Our own work using *ex-vivo* porcine conjunctival fragments has shown that these mimic more faithfully aspects of conjunctival scarring than simple 3D fibroblast-loaded collagen matrices, including physiologically relevant contraction kinetics [9]. Further work using porcine 3D cultures of mixed Tenon's capsules and bulbar conjunctival fibroblasts has shown these to be sensitive to known triggers of fibrosis (inflammatory growth factors, shear stress and aqueous humour exposure), confirming that porcine cells cultured *in vitro* can be used to model some aspects of the conjunctiva response to fibrosis [36]. However, these studies used mixed cells in moderately stiff collagen gels, which did not accurately recapitulate the Tenons' capsule/bulbar conjunctiva organisation, in which the Tenons' capsule tissue sits between the stiff sclera and the conjunctiva.

A persistent immunofibrotic reaction is a defining feature of trachoma, and in glaucoma, the presence of inflammation in the conjunctiva prior to glaucoma filtration surgery is a high risk factor for postoperative scarring and surgery failure [41]. Thus, the incorporation of an inflammatory component is crucial to better model conjunctival fibrosis. Consistent with this, we previously showed that macrophages (from normal stimulated peripheral blood monocytes or derived from human monocyte line U937) directly promote normal and fibrotic conjunctival fibroblasts contraction *in vitro* [20]. Using this model, we identified an IL6-mediated pro-fibrotic and pro-inflammatory feedback loop that could participate in trachoma progression, further demonstrating the suitability of the model to study the pathophysiology of conjunctival scarring [20]. We demonstrate here that macrophages similarly can be incorporated within the engineered conjunctiva, creating a physiologically relevant cellular environment. This will allow detailed examination of the conjunctival tissue behaviour in the context of inflammation (by varying the amount/origin of macrophages present) and fibrosis (using fibrotic fibroblasts instead of normal conjunctival cells).

Using our engineered conjunctiva, we examined the effect of potential anti-scarring drugs, including small GTPase inhibitors Ehop-016 and NSC2376 as well as doxycycline, all of which were previously shown to significantly inhibit contraction in fibroblast-populated lattices and *ex-vivo* conjunctiva contraction [9, 12, 28]. We demonstrated that our model reproduced the previously published observations and allowed multi-angle analysis of the effects of the drugs on conjunctival scarring, including macroscopic contraction, matrix degradation and cell metabolism. Furthermore, the bilayer model uniquely allows local insertion of drug delivery vehicles, and the subsequent study of drug release within a physiologically relevant tissue environment. This will be most useful for the development of slow release adjunct drug-delivery devices, or single use devices like beta-radiation which rely on modelling of penetration through tissues [42], which are critical for the prevention of postoperative scarring in low- and middle- income countries where postoperative care is limited. Indeed, our previous work on trachoma-derived fibroblasts had identified doxycycline, a widely used, broad-spectrum antibiotic, as a promising candidate to prevent scarring: doxycycline suppressed trachoma fibroblast contraction, and prevented matrix remodelling and expression of matrix metalloproteinases (MMP) previously associated with conjunctival scarring [28]. Doxycycline has also been reported to have broad anti-inflammatory properties, suggesting it could also improve the inflammation and facilitate post-surgical healing [43–46]. Consistent with this, doxycycline-loaded inserts reduced overall tissue contraction when inserted within the bilayer, providing proof of principle that the model can be useful to study the effect of locally administrated doxycycline and other anti-scarring/anti-inflammatory drugs.

Owing to the controlled extracellular environment and the use of human cells, our engineered bilayer conjunctiva model offers a cheaper, more reproducible and consistent alternative to animal models, with capacity for further optimisation to suit the research question (e.g. focusing on fibrotic environments using fibrotic fibroblasts, or on the role of inflammation by varying macrophage numbers). This would be particularly useful for studying the effect of anti-inflammatory drugs, which are currently used postoperatively to further reduce the risk of scarring and surgical failure, or pre-operatively to minimise the amount of anti-metabolite used to diminish the risk of complications [1, 35]. Such drugs (steroids, NSAIDs) have been shown to have effects on Tenon's fibroblasts, for example in classical collagen contraction models [35], and understanding their effect on the resident cells of the conjunctiva/Tenon's capsule (fibroblasts/inflammatory cells) will inform and refine the use of perioperative anti-inflammatory treatments. In addition, because potential anti-scarring drugs can be inserted fully within the 3D tissue-like environment, as it would be in patients, it represents a unique system to evaluate local drug delivery designs and their effects on the conjunctiva cellular components. Moreover, this model is easy to scale-up and to adapt to Good Manufacturing Practice by using the Real Architecture for 3D Tissue (RAFT) system [47], making it particularly suitable for screening any potential treatment, including local drug delivery, and identifying signalling pathways. This may help to minimise animal use while testing new agents.

In conclusion, we have engineered a two-layer, multi-cellular (fibroblasts and macrophages) biomimetic sub-epithelial conjunctiva, replicating the complex cellular and mechanical interactions at the bulbar conjunctiva/Tenon's capsule interface where scarring is initiated after glaucoma filtration surgery. This biomimetic allows us to explore subconjunctival tissue behaviour in the context of inflammation and fibrosis and, because potential anti-scarring drugs can be inserted fully within the 3D tissue-like environment, as it would be in patients, it represents a unique system to evaluate local drug delivery designs and their effects on the conjunctival cellular components.

## Supporting information

**S1 Fig. Calibration of cell density in compressed hydrogels.** Several cell concentrations of human fibroblasts (HTF9154 cell line) were tested in order to find the most tissue-like contraction pattern. Labels indicate the number of cells per μl of collagen gel before compression. Mean and standard deviation (SD), n = 3–6 gels.
(PDF)

**S2 Fig. Histology of the porcine conjunctiva.** Fresh porcine bulbar conjunctiva fragments representative of those used for the *ex-vivo* contraction assay were processed for haematoxylin & eosin staining. **A, B:** porcine bulbar conjunctiva presents a distinct epithelium with about 4 cell layers (arrow). The underlying conjunctival stroma is composed of fibrous matrix populated with immune and stromal cells, which becomes more diffuse and with numerous visible fibroblasts in deeper layers. **C, D:** immune cell infiltrates are present throughout the conjunctiva, with in some areas discrete conjunctival-associated lymphoid follicles (arrowhead), containing lymphoid cells and macrophages. A, C, 10X magnification; B, D 20X magnification.
(PDF)

**S3 Fig. Effect of doxycycline on *ex-vivo* conjunctiva contraction.** Porcine conjunctival fragments were cultured for 2 weeks, with/without 416 uM doxycycline and contraction was measured. Representative experiment, n = 1 (4 fragments each), mean +/- SEM.
(PDF)

**S4 Fig. Effect of NSC23766 on fibroblast-mediated contraction in non-compressed gels.**
NSC23766 treatment leads an initial reduction in contraction in non-compressed collagen gels
(standard fibroblast-populated lattices), with the difference decreasing with time. N = 6 gels,
one repeat, mean +/- SEM.
(PDF)

## Acknowledgments

The authors wish to thank Prof. Sir Peng T. Khaw for critical reading of the manuscript and
advice on the clinical relevance.

## Author Contributions

**Conceptualization:** Maryse Bailly.

**Data curation:** Katarzyna Kozdon, Bruna Caridi.

**Formal analysis:** Katarzyna Kozdon, Bruna Caridi.

**Funding acquisition:** Daniel G. Ezra, Maryse Bailly.

**Investigation:** Katarzyna Kozdon, Bruna Caridi, Iheukwumere Duru, Maryse Bailly.

**Methodology:** Katarzyna Kozdon, Bruna Caridi, Iheukwumere Duru, Daniel G. Ezra, James
B. Phillips, Maryse Bailly.

**Resources:** James B. Phillips.

**Supervision:** Daniel G. Ezra, Maryse Bailly.

**Writing – original draft:** Katarzyna Kozdon, Bruna Caridi.

**Writing – review & editing:** Katarzyna Kozdon, Bruna Caridi, Daniel G. Ezra, James B. Phil-
lips, Maryse Bailly.

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
