## [Decision Letter · Decision Letter 0]

2 Aug 2020

PONE-D-20-20861

A Tenon's capsule/bulbar conjunctiva interface biomimetic to model fibrosis and local drug delivery

PLOS ONE

Dear Dr. Baily,

Thank you for submitting your manuscript to PLOS ONE. After careful consideration, we feel that it has merit but does not fully meet PLOS ONE’s publication criteria as it currently stands. Therefore, we invite you to submit a revised version of the manuscript that addresses the points raised during the review process.

We look forward to receiving your revised manuscript.

Kind regards,

I-Jong Wang

Academic Editor

PLOS ONE

Journal Requirements:

2.Thank you for including the following ethics statement on the submission details page:

'Primary human Tenon’s capsule fibroblasts were isolated from donor tissue in accordance with the tenets of the Declaration of Helsinki and local ethics approval

(ETR reference: 10/H0106/57-2011ETR18 approved 18/6/2012 by the The Eye Tissue Repository Internal Ethics Committee of the Moorfields Eye Hospital Eye Tissue Repository)'

Please also include this information in the ethics statement in the Methods section of your manuscript.

3. Please provide additional details regarding participant consent. In the ethics statement in the Methods and online submission information, please ensure that you have specified (1) whether consent was informed and (2) what type you obtained (for instance, written or verbal, and if verbal, how it was documented and witnessed). If the need for consent was waived by the ethics committee, please include this information.”

Reviewers' comments:

Reviewer's Responses to Questions

**Comments to the Author**

1. Is the manuscript technically sound, and do the data support the conclusions?

Reviewer #1: Yes

Reviewer #2: Yes

2. Has the statistical analysis been performed appropriately and rigorously? 

Reviewer #1: Yes

Reviewer #2: Yes

3. Have the authors made all data underlying the findings in their manuscript fully available?

Reviewer #1: Yes

Reviewer #2: Yes

4. Is the manuscript presented in an intelligible fashion and written in standard English?

Reviewer #1: Yes

Reviewer #2: Yes

5. Review Comments to the Author

Reviewer #1: The authors have provided an interesting 3D compressed collagen gel that contained both human tenon fibroblasts and macrophages, as a model for studying ocular surface fibrosis and drug delivery. Overall, the paper is well written with clear figures. Nevertheless, there are few points needed to be further clarified.

1.Please define your targeting scarring process as a “conjunctival fibrosis” or “subconjunctival fibrosis”. Since glaucoma filtering surgery failed due to subconjunctival fibrosis that involved Tenon’s fibroblasts, it is important to clearly define the terminology and study aims.

2.Please provide histology image of your porcine conjunctiva as the supplement. Did the porcine conjunctival tissue by any chance contain Tenon’s capsule?

3.Please add drug effects (NSC23766, Ehop-016, doxycycline) on tissue contracture on porcine conjunctiva in Figure 4A to 4C. Therefore, we would better understand the similarities between the compressed gel and ex vivo modeling.

4.In your macrophage-incorporated model, in addition to the confocal images that illustrated the presence of macrophage, is there any evidence that the macrophages interact with fibroblasts in this model?

5.The authors use a loose, standard gel covering the compact, compressed gel to mimic Tenon’s capsule-conjunctival interface. However, Tenon’s capsule contains both collagen, elastin, or smooth muscle/fat tissue. It is elastic with dense collagen fibers (Ref:Park et al. Invest Ophthalmol Vis Sci 2016;57:5602-10). Please provide your rationale behind the two-layered design by using a loose gel to mimic Tenon’s capsule.

6.Line 96: references were wrongly inserted.

Reviewer #2: The authors try to developed a model of bulbar conjunctiva/Tenon’s capsule interface to realize the mechanism of local drug delivery through conjunctival tissue by combining plastic compression of collagen gels with a soft collagen-based layer and cultured conjunctival fibroblasts and macrophages which mimicking the mechanical proprieties and contraction kinetics of conjunctiva in this manuscript ‘A Tenon's capsule/bulbar conjunctiva interface biomimetic to model fibrosis and local drug delivery. The issue in this study is interesting. However, there are many major concerns about this paper. The reviewer suggests to decline the this paper if the major concerns in the paper without revision.

The major concerns are as follows:

1.Since the authors proposed the engineered construct contraction profile mimics ex vivo tissue contraction, making it suitable for examining such aspects of scarring and fibrosis as cell motility, matrix remodelling and degradation acting as a conjunctiva biomimetic. The reviewer’s suggestion is ‘ Please provide and compare the histologic figures between the engineered tissue and porcine conjunctival tissue. The external images in the figure1 is not significant data.”

2.The figure2 seems did not show the similar pattern as the authors mentioned “ cell -seeded compressed hydrogels and porcine conjunctiva display a similar architecture.”

3.It is important to make clear as if the similar structure. Please compare multicellular-bilayer engineered tissue and porcine tissue.

4.It is not persuasive that the resulting bi-cellular compressed tissues displayed contraction profiles (Fig. 5A) and tissue architecture (Fig. 5B) similar to those of porcine tissue (compare to Fig. 1B and 2A).

5.The authors propose that the multicellular-bilayer engineered tissue will be useful to study complex biological aspects of scarring and fibrosis with potentially significant implications for the management of scarring following glaucoma filtration surgery and other anterior ocular segment scarring conditions. However, it is not co-related to this study .

6.It uniquely allows the evaluation of new means of local drug delivery within a physiologically relevant tissue mimetic, mimicking intraoperative drug delivery in vivo.

7. The authors proposed this model is suitable for the screening of drugs targeting

scarring an inflammation, and amenable to the study of local drug delivery

devices that can be inserted in between the two layers of the biomimetic. In this paper, it is difficult to figure out the relationship.

8.Figure 6 A-B only show single layer and does not present a clear bilayer dual tissue structure. The authors should present 3D structure of multicellular-bilayer engineered tissue.

9.In the Figure 8., It is difficult to understand how can the bilayer gels be used to delivery test drug and how to test drug effect ?

6. PLOS authors have the option to publish the peer review history of their article (what does this mean?). If published, this will include your full peer review and any attached files.

Reviewer #1: No

Reviewer #2: No

---

## [Author Response · Author response to Decision Letter 0]

2 Sep 2020

PONE-D-20-20861

A Tenon's capsule/bulbar conjunctiva interface biomimetic to model fibrosis and local drug delivery

Response to the reviewers

We thank the editor and the reviewers for their comments and provide a point by point answer below (comments in bold, answer in regular font). 

Journal Requirements: (changes highlighted in blue in the marked revised manuscript)

The manuscript has been amended to meet the PLOS style requirements

2. Thank you for including the following ethics statement on the submission details page:

'Primary human Tenon’s capsule fibroblasts were isolated from donor tissue in accordance with the tenets of the Declaration of Helsinki and local ethics approval

(ETR reference: 10/H0106/57-2011ETR18 approved 18/6/2012 by the The Eye Tissue Repository Internal Ethics Committee of the Moorfields Eye Hospital Eye Tissue Repository)'. Please also include this information in the ethics statement in the Methods section of your manuscript.

The statement has been modified to include the “participant“ consent and included in the Material and Methods section (Cells and Tissue- page 10 line 120)

3. Please provide additional details regarding participant consent. In the ethics statement in the Methods and online submission information, please ensure that you have specified (1) whether consent was informed and (2) what type you obtained (for instance, written or verbal, and if verbal, how it was documented and witnessed). If the need for consent was waived by the ethics committee, please include this information.”

See above. Please note that the tissue used was not from “participants” in a study but from post mortem tissue recovered from anonymous donors. All tissues were consented for research (and transplant), with most tissue was obtained from discarded tissue after corneal transplant or from tissue unsuitable for transplant. 

Reviewers' comments: (changes highlighted in yellow in the revised manuscript)

Reviewer #1: The authors have provided an interesting 3D compressed collagen gel that contained both human tenon fibroblasts and macrophages, as a model for studying ocular surface fibrosis and drug delivery. Overall, the paper is well written with clear figures. Nevertheless, there are few points needed to be further clarified.

1.Please define your targeting scarring process as a “conjunctival fibrosis” or “subconjunctival fibrosis”. Since glaucoma filtering surgery failed due to subconjunctival fibrosis that involved Tenon’s fibroblasts, it is important to clearly define the terminology and study aims.

Both conjunctival and subconjunctival fibrosis occur in glaucoma filtration surgery (Yamanaka et al. “Pathobiology of wound healing after glaucoma filtration surgery”, BMC Ophthalmology 2015), and it is clear that the top conjunctival layer does contract in line with the subconjunctival layers, as the top tissue does not “wrinkle” during bleb failure (Prof. Sir Peng T. Khaw, personal communication). However, as rightly stated by this reviewer, subconjunctival scarring, including Tenon’s capsule contraction, is the main pathological process reducing the effectiveness of glaucoma filtration surgery postoperatively. Our model does not include a fully reconstructed conjunctival top layer (which would comprise an epithelial layer), and as such is more relevant to sub-epithelial fibrosis overall. Critically, what we aimed to reconstruct is the interface between the layers within the sub-epithelial conjunctiva (Tenons’s capsule and sub-epithelial conjunctival layer, including conjunctiva stroma) to create a model that could be used to evaluate local drug delivery within a 3D tissue-like environment. We used “conjunctival fibrosis” as a general term for the tissue, but as suggested by this reviewer, subconjunctival fibrosis is probably a more relevant and accurate terminology. We have amended this in the revised manuscript using either the term sub-epithelial or subconjunctiva as most suited (page 3, lines 37 & 39; page 7, line 111; page 19, line 367 & 370; page 26, lines 539 & 541). 

2.Please provide histology image of your porcine conjunctiva as the supplement. Did the porcine conjunctival tissue by any chance contain Tenon’s capsule?

An example histology section of the porcine conjunctiva fragments used in our study is provided in the revised supplementary figure 2 (S2 Fig, page 14, line 254). Consistent with previously described porcine bulbar conjunctiva (Crespo-Moral M et al (2020) “Histological and immunohistochemical characterization of the porcine ocular surface”. PLoS ONE 15(1): e0227732. https://doi.org/10.1371/journal.pone.0227732), the fragments had 4 epithelial cell layers with very few goblet cells. The subconjunctival layer immediately underneath the epithelium is a denser mesh of matrix and a few cells, with the deeper subconjunctival layer displaying looser tissue arrangement, with numerous cells. The Tenons’ layer is not readily identifiable in such fragments and because of lab closure due to the pandemic we were unable to process new tissue for full ocular surface histology. 

3.Please add drug effects (NSC23766, Ehop-016, doxycycline) on tissue contracture on porcine conjunctiva in Figure 4A to 4C. Therefore, we would better understand the similarities between the compressed gel and ex vivo modeling.

The effect of NSC23766 on pig conjunctiva and uncompressed collagen gels was previously described (Tovell et al, Investigative Ophthalmology & Visual Science July 2012, Vol.53, 4682-4691. doi: https://doi.org/10.1167/iovs.11-8577). This has been added on page 16, lines 303 &307. The effect of doxycycline on conjunctival tissue contraction is now presented in a new supplementary figure 3 (S3 Fig, revised manuscript page 16, line 304). Because of lab closure due to the pandemic we were unable to repeat these experiments with Ehop-016. Besides, considering the high level of toxicity displayed in the engineered tissues following treatment with Ehop-016, the value of testing the effect of that drug on conjunctival fragments is questionable. 

4. In your macrophage-incorporated model, in addition to the confocal images that illustrated the presence of macrophage, is there any evidence that the macrophages interact with fibroblasts in this model?

We have shown that macrophages influence ocular fibroblast behaviour, and in particular fibrotic features, in similar 3D co-cultures (Kechagia et al, Scientific Reports 2016, DOI: 10.1038/srep28261; Yang et al, Scientific Reports 2019, DOI: 10.1038/s41598-019-46075-1), and we have preliminary evidence that macrophages and conjunctival/Tenon’s capsule fibroblasts interact in soft collagen gels (Sharma G, et al. Invest Ophthalmol & Vis Sci. 2015;56(7):927). Further work is in progress to fully characterize these interactions in the bi-layer biomimetic. 

5.The authors use a loose, standard gel covering the compact, compressed gel to mimic Tenon’s capsule-conjunctival interface. However, Tenon’s capsule contains both collagen, elastin, or smooth muscle/fat tissue. It is elastic with dense collagen fibers (Ref:Park et al. Invest Ophthalmol Vis Sci 2016;57:5602-10). Please provide your rationale behind the two-layered design by using a loose gel to mimic Tenon’s capsule.

Historically, soft fibroblast-populated collagen matrices have been used to assess Tenon’s capsule fibroblast contraction and screen potential anti-scarring treatments (Cordeiro et al, IOVS, 2000; Daniels et al, IOVS 2003, DOI: 10.1167/iovs.02-0412 ; Yu-Wai-Man et al, Scientific Rep. 2017, DOI: 10.1038/s41598-017-00212-w). The rationale behind this is that in humans, during glaucoma filtration surgery, the Tenon’s capsule appears as a loose, soft tissue, while the overlaying subconjunctival and conjunctival tissues feel like a more discrete and stiffer layer (Prof. Sir Peng T. Khaw, personal communication). Clinically, the layers are clearly identifiable, with the soft and “floppy” Tenon’s layer residing in between the episclera (adhering to the sclera), and the conjunctiva/sub-epithelial stroma, being a more formed and stiffer layer on top. Although the Tenon’s capsule may appear as a dense layer in two-photon microscopy because of its high content in elastin and clumps of collagen fibers, it is still a loose arrangement of fibers, making the tissue overall soft and elastic. We have focused in our model -by definition a simplified version of the actual tissue- on recreating what we identified as the clinically relevant features of the subconjunctival tissue, i.e. the layered structure with differences in tissue stiffness (as perceived during the surgery) and the presence of an immune component. We have clarified this in the revised manuscript (page 7, line 111)

6.Line 96: references were wrongly inserted.

The references have been moved to the part of the sentence referring to the animal models (page 7, line 99)

Reviewer #2: The authors try to developed a model of bulbar conjunctiva/Tenon’s capsule interface to realize the mechanism of local drug delivery through conjunctival tissue by combining plastic compression of collagen gels with a soft collagen-based layer and cultured conjunctival fibroblasts and macrophages which mimicking the mechanical proprieties and contraction kinetics of conjunctiva in this manuscript ‘A Tenon's capsule/bulbar conjunctiva interface biomimetic to model fibrosis and local drug delivery. The issue in this study is interesting. However, there are many major concerns about this paper. The reviewer suggests to decline the this paper if the major concerns in the paper without revision.

The major concerns are as follows:

1.Since the authors proposed the engineered construct contraction profile mimics ex vivo tissue contraction, making it suitable for examining such aspects of scarring and fibrosis as cell motility, matrix remodelling and degradation acting as a conjunctiva biomimetic. The reviewer’s suggestion is ‘ Please provide and compare the histologic figures between the engineered tissue and porcine conjunctival tissue. The external images in the figure1 is not significant data.”

The compressed collagen gels and the conjunctival fragments detailed morphology are presented in Figure 2. Confocal microscopy is more suitable for such comparison as structures in gels made out of pure collagen are not readily visible using standard tissue section staining (such as H&E).

2.The figure2 seems did not show the similar pattern as the authors mentioned “ cell -seeded compressed hydrogels and porcine conjunctiva display a similar architecture.”

The image demonstrates similar amount of cells embedded in the 3 dimensional structures, with alternate areas of dense and loose matrix. The gels obviously can not be exactly the same as the intact conjunctiva fragments as compressed layers are made of pure collagen polymerized in vitro, whilst the porcine subconjunctival tissue matrix is rich in both large collagen fibers and elastin as rightly pointed out by reviewer 1. In addition, in Figure 2, the compressed gel shown contains only fibroblasts, and no immune cells, which are present in the conjunctival fragments. Our aim was to recapitulate the cellular components and structural/mechanical aspect of the subconjunctiva, which is what Figure 2 shows. In that sense, we believe the pattern is similar in both compressed gels and tissue fragments. 

3.It is important to make clear as if the similar structure. Please compare multicellular-bilayer engineered tissue and porcine tissue.

Please see answer above. Figure 5B shows a multicellular single-layer. Figure 6B and 6E show a multicellular bilayer.

4.It is not persuasive that the resulting bi-cellular compressed tissues displayed contraction profiles (Fig. 5A) and tissue architecture (Fig. 5B) similar to those of porcine tissue (compare to Fig. 1B and 2A).

We have updated figure panels 5A and 6F by adding conjunctiva contraction in the same graph as a direct comparison (see revised Fig5 and Fig6). Early (days 3-9) contraction profile is indeed a bit higher in conjunctiva than in the compressed single-layer gels (revised Fig 5A). However, the contraction patterns converge and are not significantly different after day 9, which is the more clinically relevant stage. Addition of the second gel layer brings the contraction pattern even closer to porcine conjunctiva (revised Fig 6A).

5.The authors propose that the multicellular-bilayer engineered tissue will be useful to study complex biological aspects of scarring and fibrosis with potentially significant implications for the management of scarring following glaucoma filtration surgery and other anterior ocular segment scarring conditions. However, it is not co-related to this study.

We respectfully do not understand this question. This model will be useful to study complex biological aspect of the scarring (such as the interactions between fibroblasts and immune cells in a physiologically relevant environment, as pertinently pointed out by reviewer 1), as well as the study of how local drug delivery might affect the contraction process. 

6.It uniquely allows the evaluation of new means of local drug delivery within a physiologically relevant tissue mimetic, mimicking intraoperative drug delivery in vivo.

Please see answer above. The ability of inserting a drug delivery device within layers the engineered tissue and to follow the drug delivery within the tissue and its effect on cells, effectively mimics what would be suitable to do to prevent scarring following glaucoma surgery.

7. The authors proposed this model is suitable for the screening of drugs targeting

scarring an inflammation, and amenable to the study of local drug delivery

devices that can be inserted in between the two layers of the biomimetic. In this paper, it is difficult to figure out the relationship.

Please see answers to points 5 and 6 above. Insertion of a drug delivery vehicle is demonstrated in figures 6E and 8.

8.Figure 6 A-B only show single layer and does not present a clear bilayer dual tissue structure. The authors should present 3D structure of multicellular-bilayer engineered tissue.

Figure 6A-B shows two layers. As explained in the text (page 19, lines 373-380), it is not possible to visualise both layers at the same time using second harmonic generation microscopy, as the collagen densities are different and require different microscope settings. Therefore, the compressed layer is readily seen in the image, whereas the soft layer is where cells appear to “float” in the air. 

9.In the Figure 8., It is difficult to understand how can the bilayer gels be used to delivery test drug and how to test drug effect ?

As demonstrated in Fig 8, the test drug can be delivered in between the two layers, effectively mimicking where one would want to insert a drug delivery device to prevent scarring at the time of surgery, and presenting proof of principle that the model is robust enough to evaluate the effect of the drug release in that context (i.e. even a large inserted device does not affect the contraction unless loaded with drug). We demonstrated this using an empty gel loaded with doxycycline, but any type of microscopic drug delivery device (e.g. biodegradable particles) can be inserted at the time of gel making or injected in between the layers. This is addressed in the discussion (page 25, lines 503-503)

---

## [Decision Letter · Decision Letter 1]

19 Oct 2020

A Tenon's capsule/bulbar conjunctiva interface biomimetic to model fibrosis and local drug delivery

PONE-D-20-20861R1

Dear Dr. Bailly,

We’re pleased to inform you that your manuscript has been judged scientifically suitable for publication and will be formally accepted for publication once it meets all outstanding technical requirements.

Kind regards,

I-Jong Wang

Academic Editor

PLOS ONE

Additional Editor Comments (optional):

Reviewers' comments:

Reviewer's Responses to Questions

**Comments to the Author**

1. If the authors have adequately addressed your comments raised in a previous round of review and you feel that this manuscript is now acceptable for publication, you may indicate that here to bypass the “Comments to the Author” section, enter your conflict of interest statement in the “Confidential to Editor” section, and submit your "Accept" recommendation.

Reviewer #1: All comments have been addressed

Reviewer #2: All comments have been addressed

2. Is the manuscript technically sound, and do the data support the conclusions?

Reviewer #1: Yes

Reviewer #2: Yes

3. Has the statistical analysis been performed appropriately and rigorously? 

Reviewer #1: Yes

Reviewer #2: Yes

4. Have the authors made all data underlying the findings in their manuscript fully available?

Reviewer #1: Yes

Reviewer #2: Yes

5. Is the manuscript presented in an intelligible fashion and written in standard English?

Reviewer #1: Yes

Reviewer #2: Yes

6. Review Comments to the Author

Reviewer #1: The authors have clarified that their model mimics subconjunctival fibrosis, and added supplementary figure that demonstrated the histology of porcine conjunctival fragments used for comparison. They also provided supplementary information of anti-scarring drug effects on porcine conjunctiva/non-compressed gel, which imitated the drug effects on current compressed gel model. Thus, the revised manuscript has answered the questions I raised for their previous version and is now considered acceptable for your journal.

Reviewer #2: (No Response)

7. PLOS authors have the option to publish the peer review history of their article (what does this mean?). If published, this will include your full peer review and any attached files.

Reviewer #1: No

Reviewer #2: No

---

## [Editor Report · Acceptance letter]

23 Oct 2020

PONE-D-20-20861R1 

A Tenon’s capsule/bulbar conjunctiva interface biomimetic to model fibrosis and local drug delivery 

Dear Dr. Bailly:

I'm pleased to inform you that your manuscript has been deemed suitable for publication in PLOS ONE. Congratulations! Your manuscript is now with our production department. 

Kind regards, 

on behalf of

Dr. I-Jong Wang 

Academic Editor

PLOS ONE